# Two Gut Microbiota-Derived Toxins Are Closely Associated with Cardiovascular Diseases: A Review

**DOI:** 10.3390/toxins13050297

**Published:** 2021-04-22

**Authors:** Tomoya Yamashita, Naofumi Yoshida, Takuo Emoto, Yoshihiro Saito, Ken-ichi Hirata

**Affiliations:** Division of Cardiovascular Medicine, Department of Internal Medicine, Kobe University Graduate School of Medicine, 7-5-1 Kusunoki-cho, Chuo-ku, Kobe 670-0017, Japan; kouchiboy@hotmail.com (N.Y.); taku_emoto@icloud.com (T.E.); y.saito.881108@gmail.com (Y.S.); hiratak@med.kobe-u.ac.jp (K.H.)

**Keywords:** cardiovascular disease, gut microbiota, lipopolysaccharide, coronary artery disease, trimethylamine N-oxide, heart failure

## Abstract

Cardiovascular diseases (CVDs) have become a major health problem because of the associated high morbidity and mortality rates observed in affected patients. Gut microbiota has recently been implicated as a novel endocrine organ that plays critical roles in the regulation of cardiometabolic and renal functions of the host via the production of bioactive metabolites. This review investigated the evidence from several clinical and experimental studies that indicated an association between the gut microbiota-derived toxins and CVDs. We mainly focused on the pro-inflammatory gut microbiota-derived toxins, namely lipopolysaccharides, derived from Gram-negative bacteria, and trimethylamine N-oxide and described the present status of research in association with these toxins, including our previous research findings. Several clinical studies aimed at exploring the effectiveness of reducing the levels of these toxins to inhibit cardiovascular events are currently under investigation or in the planning stages. We believe that some of the methods discussed in this review to eliminate or reduce the levels of such toxins in the body could be clinically applied to prevent CVDs in the near future.

## 1. Introduction

Cardiovascular diseases (CVDs) are the leading cause of death globally; an estimated 17.9 million people died from CVDs in 2016, representing 31% of all global deaths, and 85% of these deaths were because of heart attack and stroke derived from atherosclerosis (World Health Organization. Available online: https://www.who.int/en/news-room/fact-sheets/detail/cardiovascular-diseases-(cvds) (accessed on 14 March 2021)). Atherosclerosis, the precursor of coronary artery disease and stroke, is associated with several risk factors, such as dyslipidemia, diabetes mellitus, hypertension, and smoking. It can be caused by multiple factors, including metabolic abnormalities, aging, endothelial dysfunction, oxidative stress, and chronic inflammation. Currently, atherosclerotic CVDs are considered to be associated with chronic inflammatory states and their causal backgrounds, which could be used to identify novel therapeutic targets to prevent cardiovascular events [1]. 

The gut is not the first organ that was used to study the pathophysiology of CVDs. It is not only an organ that handles the digestion and absorption of food but also the largest immunologically active organ in the body that regulates the differentiation of immune cells and protects the host from invading microorganisms, with the help of native microbes that collectively form the gut microbiota. Although these microorganisms may cause infectious diseases in the host, they also perform beneficial functions, including nutrient absorption and vitamin production. Recent studies have demonstrated that gut microbiota produce and release many metabolites and toxins, some of which are absorbed into the host systemic circulation and serve as the mediators of microbial influence on the host. Therefore, the gut microbiota functions as a virtual endocrine organ that communicates with the distal organs of the host through metabolism- and immune-dependent pathways [2]. Furthermore, the gut microbiota has been linked to many diseases, including metabolic diseases, gastrointestinal diseases, cancers, and CVDs. 

Recent studies in human populations and animal models have shown that alterations in the gut microbiota might be associated with the incidence of CVDs through the production of toxins [3]. The specific characteristics of gut microbiota in coronary artery diseases have already been reported in several studies, including ours [4,5], and this kind of accumulated research could help to unravel the association between the gut microbiota and CVDs in the near future. Here, we focus on the gut microbiota-derived toxins, lipopolysaccharide (LPS) and trimethylamine N-oxide (TMAO), and summarize the role of gut microbiota in the pathophysiology of CVDs.

## 2. Lipopolysaccharides (LPS)

### 2.1. The Structure and Source of LPS

LPS, also known as endotoxin, is a constituent of the outer membrane of Gram-negative bacteria that mainly exist in the gut and oral cavity of the human body. The basic chemical structure of LPS consists of a hydrophilic region with sugar moieties bound to a hydrophobic region, known as lipid A (Figure 1A). The hydrophilic region of LPS is composed of inner and outer cores and O-antigens (species-specific repeating oligosaccharide subunits), and this region has a minimal effect on the pro-inflammatory activity of LPS (Figure 1A). The hydrophobic region, lipid A, is structurally conserved among various Gram-negative bacteria and consists of a phosphorylated diglucosamine backbone with four to seven attached acyl chains (Figure 1B). Lipid A, the ligand of toll-like receptor 4 (TLR4) and the most critical “toxin” portion of LPS, activates the innate immune system, including monocytes and macrophages, and provokes the inflammatory reaction in the host.

One question that always arises in this context is: from where does the LPS in the blood of the host come from? Currently, not much is known about this or the mechanism underlying this phenomenon. 

Considering that LPS is the major component of the outer membrane of Gram-negative bacteria, the gut microbiota could be a major source of blood LPS as several hundred trillion bacteria are present in the human gastrointestinal tract. Indeed, fecal LPS levels, reflecting gut microbiota-derived LPS, vary among individuals because of differences in gut microbial composition [6,7,8,9,10]. Increased gut permeability enhances the penetration of gut microbiota-derived LPS from the intestine into the bloodstream [11]. A high-fat diet increases the gut permeability. Akkermansia strengthens the tight junctions of the gut and prevents metabolic endotoxemia. Short-chain fatty acids produced by gut microbiota have also been reported to protect the intestinal barrier function [12]. 

In addition to gut microbiota, food is a natural source of LPS, as food and water always contain small amounts of LPS. As chylomicrons promote the intestinal absorption of LPS [13], dietary patterns reflect the LPS levels in the blood [14]. Specifically, healthy dietary food choices, including fish, fresh vegetables, and fruits, may be associated with positive health outcomes, as they help to reduce endotoxemia. However, the intake of energy, fiber-rich food, individual macronutrients, or fatty acids does not affect the blood LPS levels. Circulating LPS levels are inversely associated with adherence to a Mediterranean diet, particularly with fruit and legume intake [15]. Furthermore, the oral intake of probiotic products containing LPS or Gram-negative bacteria does not pose a health risk [16]. The association between food and blood LPS levels was evaluated using in vivo experiments. Mice that were orally administered LPS diluted in oil showed increased blood LPS levels [17]. These data suggested that food was a source of blood LPS. However, further in vivo studies are needed to know whether dietary intake directly determines the blood LPS levels and check whether reducing the blood LPS levels by modulating the dietary habits of patients can be feasible. 

CVDs have been associated with several traditional risk factors, such as hypertension, dyslipidemia, diabetes, smoking, and obesity [18]. Inflammatory processes are also known to play crucial roles in the development of CVDs and the complications associated with these diseases [19,20,21,22]. LPS, a well-known inflammatory substance, is vital to maintain the structural and functional integrity of the outer membrane [23] of Gram-negative bacteria. As the stimulation of TLR4 by LPS induces the release of critical pro-inflammatory cytokines that are necessary to activate potent immune responses [24], many clinical studies have explored the association between the LPS levels and disease progression. Large epidemiologic evidence shows that endotoxemia, which indicates high levels of LPS in the blood, is a strong risk factor for atherogenesis and acts as a link in the association between LPS and atherosclerotic disease [25]. Increased concentrations of plasma LPS and cytokines have been reported in patients with chronic heart failure during acute edematous exacerbation, suggesting that endotoxins trigger the activation of immune system in patients with chronic heart failure [26]. Furthermore, plasma LPS levels are predictive of major adverse cardiovascular events in patients with atrial fibrillation [15]. Endotoxemia is also involved in obesity and insulin resistance mechanisms [17,27,28], which are closely related to CVDs. These clinical and basic research studies clearly indicate that endotoxemia is associated with CVDs, and it could serve as a powerful therapeutic target in the treatment of CVDs.

Recently, we reported that patients with CVDs have higher fecal LPS levels and associated risk factors than the control patients without CVDs. Although, in our previous study, we could not show a strong positive correlation between fecal LPS levels and plasma LPS levels in patients with CVD, fecal LPS levels are now considered to be potent elements that are used to explain the correlation between the gut microbiota and incidence of CVDs. More interestingly, the structures of lipid A moieties of LPS differ among different bacterial species [29] (Figure 1B). These structural differences may be the main factors that determine LPS activity [6]. For example, *Bacteroides* is known to have tetra- and penta-acylated lipid A moieties, whereas *Escherichia* has hexa-acylated lipid A moiety [29,30]. Generally, the tetra- and penta-acylated lipid A moieties elicit reduced TLR4 responses compared to the hexa-acylated lipid A moiety [30]. This indicates that both the gut bacterial composition and the type of lipid A moieties these bacteria possess may be important factors that affect the association between gut microbial LPS and CVDs.

### 2.2. Bacteroides Administration Reduces the LPS Activity and Inhibits Atherosclerosis

*Bacteroides* spp. are the dominant bacteria seen in the human gut and have tetra- and penta-acylated lipid A moieties [29]. Our previous analysis using fecal 16S rRNA gene sequencing revealed a significantly decreased abundance of *Bacteroides* spp., especially *Bacteroides vulgatus* and *B. dorei*, in patients with coronary artery disease [4,7]. Therefore, we administered *B. vulgatus* and *B. dorei* to atherosclerosis-prone apolipoprotein E-deficient mice to clarify the causality and found that gavage with these two strains altered the gut microbial composition and attenuated atherosclerotic lesion formation, thus markedly ameliorating endotoxemia. We also predicted the gut bacterial gene functions based on 16S rRNA gene sequences using the Phylogenetic Investigation of Communities by Reconstruction of Unobserved States (PICRUSt) and found that the expression levels of genes involved in lipid A biosynthesis were significantly decreased in mice that were administered *Bacteroides*. Therefore, we considered the fecal LPS levels as indicators of LPS produced by gut microbiota and found that fecal LPS activity was significantly lower in mice administered with *Bacteroides* than in the control mice. In summary, gavage with *B. vulgatus* and *B. dorei* decreased the fecal and plasma LPS levels and protected against atherosclerosis in mice.

Furthermore, we assessed the LPS activity based on the structure of lipid A moiety. We extracted lipid A from the two *Bacteroides* strains, *B. vulgatus* and *B. dorei*, to examine its biological activity and compared it with the LPS in *Escherichia coli*, which exhibits a strong pro-inflammatory activity. We found that the LPS in *Bacteroides* induced relatively low levels of pro-inflammatory cytokine production in cultured monocyte cell lines in vitro [6]. Furthermore, the endotoxin units of the *Bacteroides* LPS, as determined by using limulus amoebocyte tests, were significantly lower than those of the *E. coli* LPS [6]. Our findings suggest that *Bacteroides* administration may serve as a novel and effective therapeutic strategy for suppressing the inflammatory responses in patients with CVDs and pave the way for further studies investigating the effects of fecal LPS levels on the prevention of CVDs.

### 2.3. Future Perspective on the Clinical Applications of LPS

Accumulating evidence suggests that systemic endotoxemia and gut microbiota-derived LPS are involved in the onset and progression of CVDs and many other prevalent disorders, such as the inflammatory bowel disease, obesity and related metabolic diseases, and non-alcoholic steatohepatitis [6,7,31,32,33]. LPS is considered to be a toxin in inflammatory diseases, including CVDs, and is involved in the pathophysiology of the onset and progression of CVDs. So far, therapies that directly decrease the blood or fecal LPS levels do not exist. Thus, the development of clinical applications for managing endotoxemia or modulating the gut microbial composition can be novel therapeutic options for the treatment of CVDs.

## 3. Trimethylamine N-Oxide (TMAO)

### 3.1. TMAO as a Risk Factor for Cardiovascular Diseases

In 2011, Dr. Hazen and his colleagues used a metabolomics approach to make a remarkable discovery in the study of atherosclerosis and revealed that the gut-derived metabolite, TMAO, is an independent predictor of cardiovascular events in a large clinical cohort of patients with CVDs. They also reported that dietary supplementation with choline or TMAO promoted atherosclerosis via the upregulation of multiple macrophage scavenger receptors in atherosclerosis-prone mice [34]. Phosphatidylcholine, a dietary component found in food sources, such as cheese, egg yolk, meat, and shellfish, is converted to choline in the gut and subsequently metabolized to trimethylamine (TMA) using the gut microbial enzyme, TMA-lyase. TMA is absorbed from the gut into the portal circulation and then converted to TMAO via flavin-containing monooxygenases, host enzymes, in the liver [34]. They also demonstrated that dietary L-carnitine, an abundant nutrient in red meat containing a trimethylamine structure similar to that of choline, contributes to the elevation of plasma TMAO levels and accelerates atherosclerosis [35]. TMAO is elevated in CVD patients with coronary artery disease, thrombosis [36], chronic kidney disease (CKD) [37], and heart failure (HF) [38] and is associated with adverse cardiovascular events and all-cause mortality [39] (Figure 2A). Elevated TMAO levels are reported to be strongly associated with the degree of renal function and increased systemic inflammation in CKD patients and TMAO acts as an independent predictor of mortality in this cohort of patients with severe CKD [40]. It is reasonable to conclude that TMAO acts as a gut microbiota-derived uremic or cardiovascular toxin that contributes to systemic inflammation.

### 3.2. The Composition of Gut Microbiota and Plasma TMAO Levels in HF Patients 

The number of elderly patients with HF, especially HF with preserved ejection fraction (HFpEF), has increased with the rapid increase in aging population in developed countries. Because of the limitations of finding effective drugs for HFpEF, it is necessary to find effective interventions in the nutrition and lifestyles of patients with HF. Alterations in gut microbial composition have already been described in patients with HF, especially those with reduced microbial diversity and facing depletion of core gut microbiota [41,42,43]. However, the association between the composition of gut microbiome and HF pathophysiology remains unclear. We suspect that the heterogeneity of HF makes it difficult to focus on specific gut microbiota from the data of comparison between the control patients and patients with HF and that intestinal edema caused by HF affects the composition of gut microbiota. Therefore, we assessed the composition of gut microbiota and plasma-related metabolites in patients with decompensated (Decomp) HF when they were admitted to the hospital and the compensated (Comp) phase in the same patients after HF treatment [44].

Plasma TMAO concentrations were found to be elevated in both Decomp and Comp HF patient groups (Figure 2B), which suggested that TMAO elevation was neither temporal nor a therapeutic target after hospitalization to prevent the next event of heart failure. At the phylum level, levels of Actinobacteria were increased in HF patients compared to those in the control patients (Figure 2C). At the genus level, Bifidobacterium was abundant in HF patients, although it was difficult to assess how this change in gut microbiota affected the plasma TMAO levels using amplicon sequencing of the bacterial 16S ribosomal RNA gene [44]. The three key microbial functional gene clusters involved in the production of TMA are choline TMA-lyase (cutC) and its activator (cutD) (cutC/D), a glycyl radical enzyme and a glycyl radical-activating protein [45]; carnitine oxygenase A/B (cntA/B), a two-component Riesketype oxygenase/reductase complex [46]; betaine reductase pathways (Figure 2A). In this review, we describe the data of shotgun metagenome sequencing analysis to elucidate which TMA generation pathway from choline, carnitine, or betaine contributes to the increase in TMAO levels in HF. We also discuss the kinds of bacteria that have cutC/D or cntA/B and contribute to the elevation of plasma TMAO levels.

### 3.3. Therapeutic Candidates to Decrease the Levels of Plasma TMAO 

From a clinical perspective, a new method or therapeutic intervention to suppress the plasma TMAO levels in patients with CVDs is desired. A natural structural analog of choline, 3,3-dimethyl-1-butanol (DMB), has been shown to non-lethally inhibit TMA production from cultured bacteria [47]. However, despite the high dose of DMB provided, choline-induced elevation in TMAO levels increased the platelet aggregation, and the shortened thrombus formation process was not fully rescued in mice. Therefore, they tried to develop second-generation TMA-lyase inhibitors with improved therapeutic potential [48]. In particular, a new choline trimethylamine lyase inhibitor, iodomethylcholine, improved the remodeling and cardiac function in a heart failure murine model with transverse aortic constriction [49].

“Leaky gut” in patients with CVDs is a mechanism to reduce the plasma TMAO levels. Abe et al. showed that linaclotide, a guanylate cyclase C agonist, decreased the plasma levels of TMAO and improved the renal function in an adenine-induced renal failure murine model by ameliorating the expression of claudin-1 in the gut [50].

Probiotics present another option to reduce the plasma TMAO levels by suppressing the TMA production in the gut via the modulation of gut microbiota, metabolomics profile, miRNA expression, or probiotic antagonistic abilities [51].

## 4. Conclusions

Mounting evidence from human and animal studies supports the idea that gut microbiota and their metabolites or toxins can influence the host health and may even cause diseases in the host. The identification of bacterial products that can modulate the physiological and pathophysiological processes of the host has opened various possibilities, particularly demonstrating that numerous microbial pathways can act as therapeutic targets to inhibit CVDs. In particular, to eliminate or reduce the levels of the detrimental toxins, LPS and TMAO can be used to develop promising and effective therapeutic strategies to reduce the number of patients with CVDs. However, we have not yet been able to demonstrate how these toxin levels are determined and regulated in the host. Thus, further studies are needed to clarify the causal relationship between the gut microbiota-derived toxins and CVDs to determine whether therapeutic intervention can be effective and identify the best method of therapeutic intervention for CVDs. We hope that future studies will be able to further explore the association between the gut microbiota and their metabolites, including toxins, in order to improve the prognosis of patients with CVDs.

## Figures and Tables

**Figure 1 toxins-13-00297-f001:**
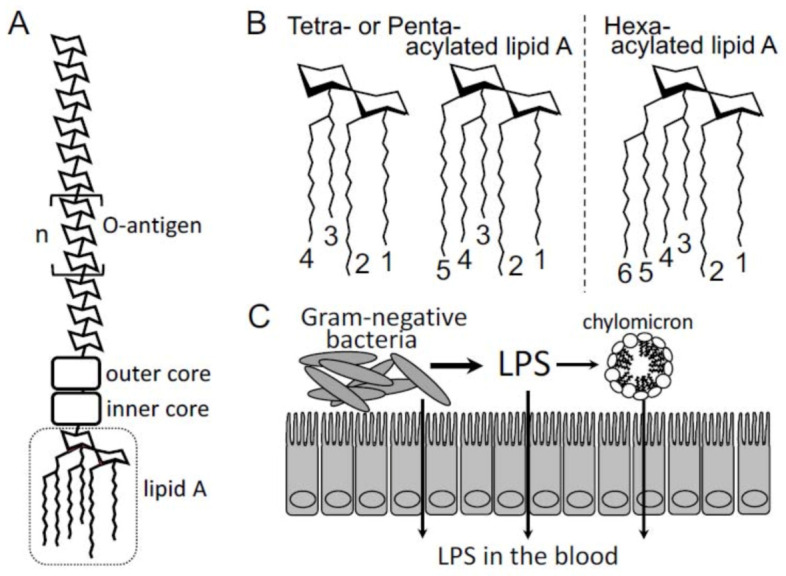
Structure of lipopolysaccharide (LPS) and lipid A. (**A**) The portion in the dotted box represents the lipid A moiety. The other portions represent inner and outer cores and the O-antigens. (**B**). Left panel represents tetra- or penta-acylated lipid A derived from the *Bacteroides* species. Right panel represents hexa-acylated lipid A derived from *Escherichia coli*. (**C**) This panel demonstrates the mechanism of lipopolysaccharide (LPS) transfer from the gut lumen into the blood vessel. The precise mechanisms have not yet been determined and these are just suggested candidate pathways.

**Figure 2 toxins-13-00297-f002:**
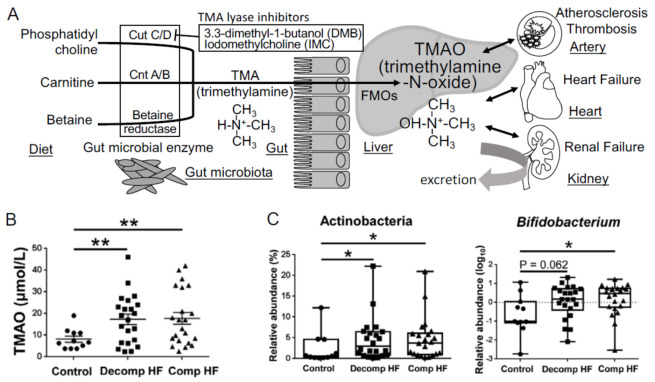
Metabolism of the gut microbiota-derived trimethylamine N-oxide (TMAO). (**A**) The metabolic pathways involved in the production of TMAO. (**B**) Plasma TMAO levels in heart failure patients (Decomp, worsening decompensated phase; Comp, compensated phase after treatment) and non-heart failure control patients. (**C**) Changes in gut microbiota involved in heart failure. Data are presented as mean ± SE. * *p* < 0.05, ** *p* < 0.01.

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
