# Peer review of "Two Gut Microbiota-Derived Toxins Are Closely Associated with Cardiovascular Diseases: A Review"

_toxins, 2021, doi:10.3390/toxins13050297_

Round 1

Reviewer 1 Report

Gut microbiota derived toxins and CVD

This is an interesting and up to date paper with good examples, some from the Authors laboratory. The paper reflects on gut-derived molecules – called toxins – that mediate CVD and even has sections on how these may be addressed therapeutically. The paper appears to be sound in its content.

With some modifications the paper should be interesting for the audience.

The English language is quite good but needs input from a native English-speaking person (or equivalent) to correct many small items, which will make the paper much easier to be read and followed. There is a little bit of non-scientific language that if removed would improve the paper. There are some spelling/typographical errors such as form/from.

A large issue is the question of context – much work on the gut microbiome has focused on its beneficial effects even to the extent of faecal transplants. Here the focus is in toxins – the paper would benefit from two additions – several extra sentences at the very start about CVD to put it in context and then a second paragraph in the Introduction, which gives a broad over view of the current state of microbiome studies and disease (CVD).

Much of the existing work has been about specific genetically characterised microbes and the changes in health, disease and treatment and some effort should be made to link toxins with individual microbes.

Title – the title is probably too broad in that it refers to “toxins” but really addresses only two toxin families – it would be more specific and useful if the 2 families were mentioned in the title.

The referencing is not solid and consistent – there are some statements of fact/data which have no references in the sentence. In addition, it would be better when mentioning specific authors to put the reference immediately with the name e.g. Smith et al. (REF).

END

Author Response

Response letter to the reviewer 1

Journal: Toxins
Manuscript ID: toxins-1173491
Previous Title: Gut microbiota-derived toxins and metabolites in cardiovascular diseases.

New Title: Gut microbiota-derived two toxins are closely associated with cardiovascular diseases: A review

              We are grateful for the insights and valuable comments from the reviewer 1. We have carefully addressed each comment and revised the manuscript. The manuscript has considerably benefited from these modifications.

The reviewer 1; This is an interesting and up to date paper with good examples, some from the Authors laboratory. The paper reflects on gut-derived molecules – called toxins – that mediate CVD and even has sections on how these may be addressed therapeutically. The paper appears to be sound in its content. With some modifications the paper should be interesting for the audience.

              Thank you for high evaluation of our manuscript. I sincerely respond to the reviewer's comment point-by-point. These changes must improve the revised manuscript.

â—‹ The English language is quite good but needs input from a native English-speaking person (or equivalent) to correct many small items, which will make the paper much easier to be read and followed. There is a little bit of non-scientific language that if removed would improve the paper. There are some spelling/typographical errors such as form/from.

              According to the reviewer's suggestion, the paper was edited by the native English user (Editage; www.editage.com). I hope these changes could improve it.   

â—‹ A large issue is the question of context – much work on the gut microbiome has focused on its beneficial effects even to the extent of faecal transplants. Here the focus is in toxins – the paper would benefit from two additions – several extra sentences at the very start about CVD to put it in context and then a second paragraph in the Introduction, which gives a broad over view of the current state of microbiome studies and disease (CVD).

              Based on the reviewer's suggestion, I added several sentences associated with general status of CVD in the first paragraph of the introduction section. And the second and third paragraphs focused on the gut microbiota, gut microbiota-derived toxins and CVD.

â—‹ Much of the existing work has been about specific genetically characterised microbes and the changes in health, disease and treatment and some effort should be made to link toxins with individual microbes.

              As the reviewer pointed out, many research paper focused and reported the specific gut flora types or specific genetically characterized microbes when examining the association with some diseases. However, the name of microbiota rarely lead to the understanding or discussing the effects of gut microbiota on host metabolic and immune function. Taken these, when focusing on the gut-microbial toxins, I think it might be better not to show many names of gut microbiota. We had already demonstrated several bacterial names of gram-negative bacteria, because those are closely associated with the strength of the toxins (LPS). We dared not to demonstrate the names of microbes having TMA lyase, because a lot of microbes have the enzyme and listing them must confuse the readers.  

â—‹Title – the title is probably too broad in that it refers to “toxins” but really addresses only two toxin families – it would be more specific and useful if the 2 families were mentioned in the title.

              I agreed the reviewer's opinion and changed the title of this manuscript. " Gut microbiota-derived toxins and metabolites in cardiovascular diseases." to " Gut microbiota-derived two toxins are closely associated with cardiovascular diseases: A review.

â—‹The referencing is not solid and consistent – there are some statements of fact/data which have no references in the sentence. In addition, it would be better when mentioning specific authors to put the reference immediately with the name e.g. Smith et al. (REF).

              I added several reference papers and changed the reference numbers in the revised manuscript.

Reviewer 2 Report

The authors reviewed the association of the gut microbiota-derived toxins, such as LPS and TMAO with CVDs.

In the paragraph starting with “Then, one primitive question is from where the blood LPS in host comes?”, the authors mentioned that food is a natural source of LPS. However, endotoxemia is mainly caused by the released LPS from the intestine to the bloodstream. Therefore, after explaining the process of gut dysbiosis, loss of gut integrity, metabolic endotoxemia, and low-grade inflammation (and other metabolic diseases), it would be better to explain foods associated with gut dysbiosis, not LPS in the food itself.   

It there any other examples of which bacteria produce tetra-, penta-, or hexa-acylated lipid A?

Minor comments

Several bacteria names were not italicized.

The citation format is not consistent.

Change “Further, The gut” to “Further, the gut” in the introduction section.

Author Response

Response letter to the reviewer 2

Journal: Toxins
Manuscript ID: toxins-1173491
Previous Title: Gut microbiota-derived toxins and metabolites in cardiovascular diseases.

New Title: Gut microbiota-derived two toxins are closely associated with cardiovascular diseases: A review

              We are grateful for the insights and valuable comments from the reviewer 2. We have carefully addressed each comment and revised the manuscript. The manuscript has considerably benefited from these modifications.

The reviewer 2; The authors reviewed the association of the gut microbiota-derived toxins, such as LPS and TMAO with CVDs.

â—‹ In the paragraph starting with “Then, one primitive question is from where the blood LPS in host comes?”, the authors mentioned that food is a natural source of LPS. However, endotoxemia is mainly caused by the released LPS from the intestine to the bloodstream. Therefore, after explaining the process of gut dysbiosis, loss of gut integrity, metabolic endotoxemia, and low-grade inflammation (and other metabolic diseases), it would be better to explain foods associated with gut dysbiosis, not LPS in the food itself.   

              On the basis of the reviewer's useful suggestion, I added the paragraph focusing on gut permeability, barrier function, and metabolic endotoxemia in association with LPS (page 2). After that, I changed the sentence of foods in association with blood LPS (pages 2~3).   

â—‹ It there any other examples of which bacteria produce tetra-, penta-, or hexa-acylated lipid A?

              Vatanen T, et al. reported that several Bacteroides species and Prevotella copri have penta- or tetra- acylated lipid A (Cell. 2016;165(4):842-53.). In those, they demonstrated that LPS derived from Bacteroides dorei or vulgatus is the weakest pro-inflammatory agonist. E.coli LPS and LPS derived from Klebsiella species have 6 acylated lipid A and can induce strong inflammation via activating Toll-like receptor 4. These are solid facts and I added a sentence associated with this paper in page 4 line 4 in the revised manuscript.                                Porphyromonas gingivalis, mainly existing in oral cavity, produces 2 different penta- or tetra- acylated lipid A. Both pent- and tetra- acylated lipid A containing LPS can induce weaker immuno-responses compared with LPS derived from E. coli. (Qiu C, et al. Front Cell Infect Microbiol. 2021;11:606986.) Further, it is also the fact that tetra- and penta-acylated lipid A structures of Porphyromonas gingivalis LPS differentially activate TLR4-mediated NF-κB signal transduction cascade and immuno-inflammatory response in human gingival fibroblasts. (Herath TD, et al. PLoS One. 2013;8(3):e58496.)  The functional differences of LPS depending on the number of lipid A have not yet clearly clarified.

â—‹ Minor comments; 1.Several bacteria names were not italicized.  2. The citation format is not consistent.  3. Change “Further, The gut” to “Further, the gut” in the introduction section.

              All were altered as shown in the revised manuscript. Based on the native language editing, many changes were made. 

Round 2

Reviewer 2 Report

The authors have addressed most of my previous comments and the manuscript is greatly improved.